# MambaTrack: a simple baseline for multiple object tracking with State Space Model

## ABSTRACT

Tracking by detection has been the prevailing paradigm in the field of Multi-object Tracking (MOT). These methods typically rely on the Kalman Filter to estimate the future locations of objects, assuming linear object motion. However, they fall short when tracking objects exhibiting nonlinear and diverse motion in scenarios like dancing and sports. In addition, there has been limited focus on utilizing learning-based motion predictors in MOT. To address these challenges, we resort to exploring data-driven motion prediction methods. Inspired by the great expectation of state space models (SSMs), such as Mamba, in long-term sequence modeling with near-linear complexity, we introduce a Mamba-based motion model named Mamba moTion Predictor (MTP). MTP is designed to model the complex motion patterns of objects like dancers and athletes. Specifically, MTP takes the spatial-temporal location dynamics of objects as input, captures the motion pattern using a bi-Mamba encoding layer, and predicts the next motion. In real-world scenarios, objects may be missed due to occlusion or motion blur, leading to premature termination of their trajectories. To tackle this challenge, we further expand the application of MTP. We employ it in an autoregressive way to compensate for missing observations by utilizing its own predictions as inputs, thereby contributing to more consistent trajectories. Our proposed tracker, MambaTrack, demonstrates advanced performance on benchmarks such as Dancetrack and SportsMOT, which are characterized by complex motion and severe occlusion.

## CCS CONCEPTS

• **Computing methodologies** → **Tracking**; *Motion capture*.

## KEYWORDS

Multiple object tracking, Nonlinear motion, Occlusion handling, Motion prediction, Mamba, State Space Model

**ACM Reference Format:**
Anonymous Author(s). 2018. MambaTrack: a simple baseline for multiple object tracking with State Space Model. In *Proceedings of Make sure to enter the correct conference title from your rights confirmation emai (Conference acronym 'XX)*. ACM, New York, NY, USA, 10 pages. https://doi.org/XXXXXXX.XXXXXXX

## 1 INTRODUCTION

Multi-object tracking (MOT) is a fundamental computer vision task aimed at locating objects of interest and associating them across video frames to form trajectories. It has extensive applications in various domains, including autonomous driving [5, 12, 43], human behavior analysis [9, 40, 49], and robotics [28]. Tracking-by-detection [3, 6, 26, 46, 52] has been the dominant paradigm due to its succinct design, which involves two main steps: 1) obtaining the bounding boxes of objects using an off-the-shelf detector, and 2) associating these detections into trajectories based on appearance or motion cues. This paradigm has seen significant progress over the past decade, particularly in scenarios [10, 30] characterized by distinguishable appearance and simple motion patterns.

Despite the commendable performance of these trackers on pedestrian tracking benchmarks [10, 30], their efficacy diminishes notably in intricate scenarios [9, 40], typified by various and rapid movements, as well as less discriminative appearances. The primary challenge encountered in DanceTrack and SportsMOT resides in the data association phase. Specifically, the limitations stem from the inefficacy of object appearance cues in distinguishing between distinct objects and the insufficiency of conventional motion predictors, Kalman filter, in accurately forecasting object positions in scenes characterized by nonlinear motion patterns and frequent occlusions.

To address the challenges posed by these complex scenarios, we turn our attention to leveraging motion information for object association. Given the unreliability of appearance cues, our emphasis is on designing a learnable motion predictor capable of capturing object motion patterns solely from object trajectory sequences. While Long Short-Term Memory (LSTM) [17] and Transformer [42] architectures are both prominent in sequence modeling, they face distinct challenges. LSTM is criticized for its inefficient training and limited capacity for long-term modeling, whereas Transformer suffers from quadratic computational complexity relative to sequence length during inference. In recent years, state space models (SSMs) have shown promise in optimizing performance and computational complexity concurrently. These models capture sequence information through convolutional computing and achieve near-linear complexity during inference. A recent advancement, Mamba [13], integrates a selective mechanism into SSMs to attend to important parts of sequence data, akin to attention mechanisms [42]. Inspired by Mamba's success in sequence data modeling, we are motivated to incorporate it into Multi-Object Tracking to capture complex object motion patterns. Therefore, we propose a learnable motion predictor, Mamba moTion Predictor (MTP), which takes the historical motion information of object trajectories as input, employs a bi-Mamba encoding layer to encode movement information and predicts the next movement of objects. Subsequently, data association is performed based on the Intersection-over-Union (IoU) similarities between the predicted bounding boxes of tracklets and

the detections of the current frame. Experimental results validate the effectiveness of MTP, particularly its significant performance dominates over the classical Kalman filter.

Despite exploiting MTP for object association between adjacent frames, we extend its usage to achieve long-term association. Specifically, to re-establish lost tracklets caused by occlusions or detector failures, we introduce a tracklet patching module. This module compensates for missing observation points by employing MTP in an auto-regressive manner, wherein it takes its own predictions as input to continue predicting the next motion of the lost tracklets. With the assistance of tracklet patching, our proposed tracker, MambaTrack, generates more consistent trajectories.

In conclusion, the major contributions of this work are as follows:

- We propose a data-driven motion predictor, Mamba moTion Predictor (MTP), designed to model diverse motion patterns in complex scenarios.
- We introduce a tracklet patching module that employs MTP in an auto-regressive manner to re-establish the lost tracklets.
- Equipped with the designed MTP and the tracklet patching module, the proposed online tracker, MambaTrack, addresses the challenging data association problem in complex dancing and sports scenarios. As a motion-based online tracker, MambaTrack achieves state-of-the-art performance on the two merging benchmarks, DanceTrack [40] and SportsMOT [9].

## 2 RELATED WORK

### 2.1 Tracking-by-detection methods

With the rapid advancement of detection and re-identification techniques [7, 11, 34, 35, 48], tracking-by-detection (TBD) methods [3, 6, 32, 44–46, 52, 53] have made significant progress. These methods utilize existing detectors to obtain detection results from video frames, which are then associated with forming object trajectories. Some TBD methods generate object trajectories using complex optimization algorithms [4, 23] in an offline manner, while others operate in an online manner, associating detections with tracklets frame-by-frame. Given the practicality of online methods, researchers have focused on enhancing them from various perspectives. For instance, methods like JDE [45] and FairMOT [53] extract object spatial locations and appearance embeddings from a shared network, thereby improving accuracy and inference efficiency. Additionally, QDTrack [32] employs a contrastive learning strategy to acquire reliable appearance cues. Moreover, ByteTrack [52] employs cascading matching strategies to handle detections with varying confidence levels obtained from a modern detector, resulting in impressive performance. However, the conventional benchmarks [10, 30] primarily feature distinct appearances and regular motion patterns, leading to a heavy reliance on appearance cues and limited utilization of motion information for data association.

### 2.2 Motion models

The changes in the spatial locations of objects serve as crucial cues for tracking objects across frames. Motion models utilized in multi-object tracking can be broadly categorized as filter-based or learning-based. The classical work, SORT [3], employs the Kalman Filter (KF) [20] to estimate the motion state of objects. Although subsequent works [45, 46, 52, 53] inherit this motion model, they are primarily designed for tracking objects with regular motion patterns and struggle in more complex motion scenarios. OC_SORT [6] addresses the inherent limitations of KF and enhances its capability to handle nonlinear motion and occlusion scenarios. Learning-based methods predict object inter-frame offsets from video frames or rely solely on trajectory information. For example, Tracktor [2] incorporates a regression branch to predict object displacements using information from two consecutive frames. CenterTrack [55] predicts the center offsets of objects using information from two consecutive frames and the last heatmap as input. ArTIST [38] treats object motion as a probability distribution and employs an MA-Net to model interactions among objects. However, these methods tend to be computationally intensive or require complex training procedures. In this work, our proposed tracker relies solely on the historical bounding box sequences of objects to predict their future locations. By adopting this approach, we aim to propose a simple motion-based tracker in diverse motion scenarios while maintaining high accuracy.

### 2.3 State Space Models

Inspired by control theory, the integration of linear state space equations with deep learning has been explored to enhance the modeling of sequential data. This fusion was initially catalyzed by the introduction of the HiPPO matrix [14], which laid the groundwork for subsequent developments. LSSL [16] represents a pioneering effort in this domain, utilizing linear state space equations to model sequence data. [15] introduces Structured State-Space Sequence S4 to model long-range dependency, which advanced the field by employing linear state space representations for contextualization, demonstrating robust performance across a spectrum of sequence modeling tasks. The inherent characteristic of linear scalability in the sequence length of SSMs attracts more attention. Furthermore, SGConv [25] offers an innovative perspective by recasting the S4 model as a global conventional framework. In pursuit of enhanced computational efficiency, GSS [29] incorporates a gating mechanism within the attention unit, thereby reducing the dimension of the state space module. A seminal contribution to the field is the introduction of the S5 layer [39], which encompasses the parallel scan and the MIMO SSM. This layer significantly streamlines the utilization and implementation of state space models, paving the way for widespread adoption. The state space model has been successfully applied in the domain of computer vision by various research initiatives, such as ViS4mer [18], S4ND [31] and TranS4mer [19].

Recently, Gu et al. [13] introduced a data-dependent SSM layer in their work, which establishes a generic language model backbone termed Mamba. Mamba exhibits superior performance compared to Transformers across various scales on extensive datasets while also benefiting from linear-time inference and efficient training procedures. Building on the success of Mamba, Mamba attracted the attention of a lot of researchers. MoE-Mamba [33] integrates a Mixture of Expert approach with Mamba, thereby unleashing the

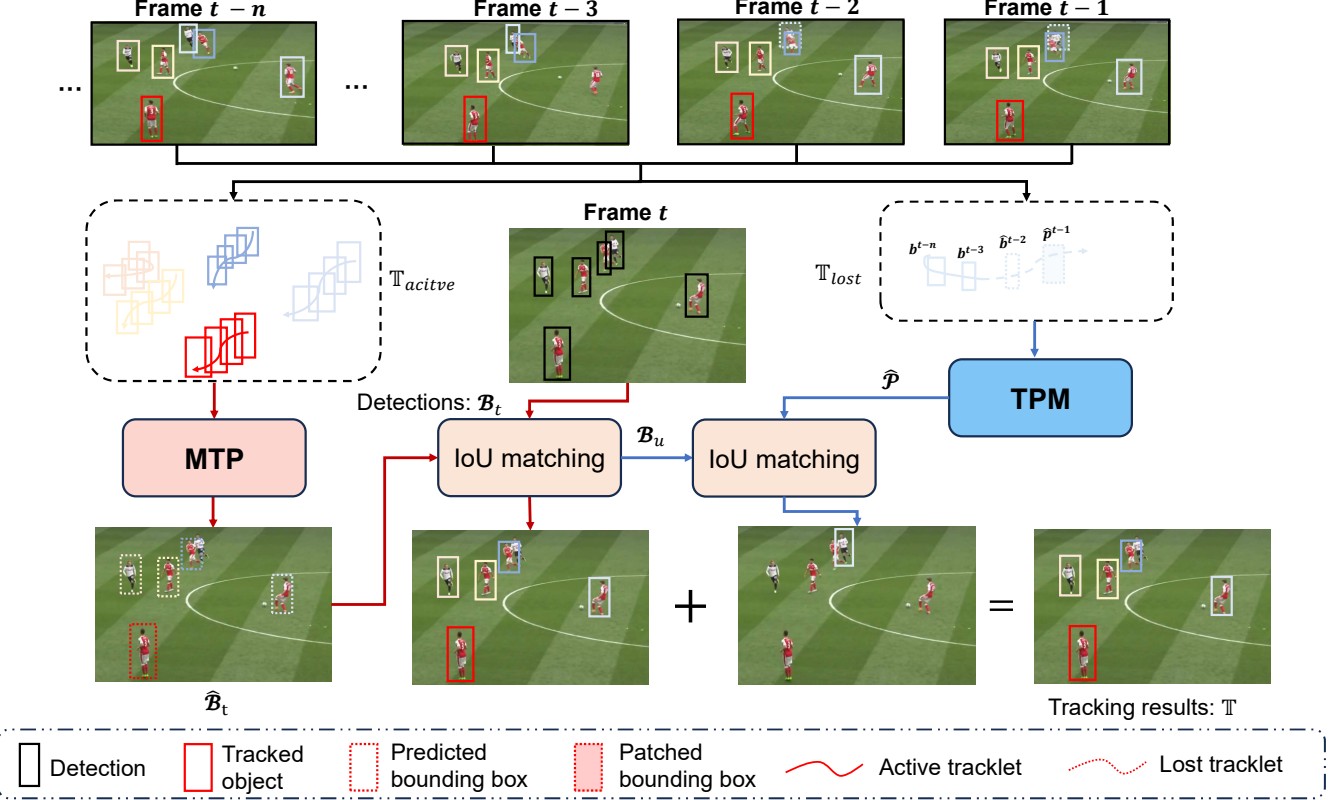

Figure 1: Overall architecture of the proposed methods. First, we employ the proposed Mamba Motion Predictor (MTP) to predict the bounding boxes $\hat{\mathcal{B}}_t$ of active tracklets in the subsequent frame. These predictions are then matched with the detection results $\mathcal{B}_t$ of the current frame $t$ based on Intersection-over-Union (IoU) similarity. Subsequently, the Tracklet Patching Module (TPM) predicts the bounding box $\hat{P}$ of lost tracklets through autoregression and pairs it with the remaining detections $B_u$. Finally, the results of the matching steps are combined to derive the tracking results $\mathbb{T}$. Different colored bounding boxes represent objects of different identities.

scalability potential of SSMs and achieving performance comparable to Transformers. The VideoMamba [24] effectively employs Mamba's linear complexity operator to facilitate efficient long-term modeling, demonstrating notable advantages in tasks related to understanding lengthy videos.

To the best of our knowledge, there has been no prior research that explores the Mamba architecture to multi-object tracking. It is intuitive to speculate that the Mamba architecture could serve as an effective and efficient solution for the MOT problem, particularly in scenarios involving occlusion and non-linear motion. Therefore, we try to incorporate the Mamba model in multi-object tracking task.

## 3 PRELIMINARIES

**State Space Models.** State Space Models (SSMs) are inspired by linear time-invariant (LTR) system that map the input sequence $x(t) \in \mathbb{R}$ to a response $y(t) \in \mathbb{R}$ through the hidden state vector $h(t) \in \mathbb{R}^N$. Mathematically, the dynamics of the system can be modeled by a set of first-order differential equations:

$$
\begin{aligned}
\dot{h}(t) &= Ah(t) + Bx(t), \\
y(t) &= Ch(t) + Dx(t).
\end{aligned}
\tag{1}
$$

where matrices $A \in \mathbb{R}^{N \times N}$ represents the evolution parameters and $B \in \mathbb{R}^{N \times 1}, C \in \mathbb{R}^{N \times 1} D \in \mathbb{R}^{N \times 1}$ are the projection parameters.

**Discretization.** Since SSMs are designed to operate on continuous signal $x(t)$, they inherently cannot process discrete sequences like time series and natural language $\{x_0, x_1, \dots\}$. Therefore, it becomes imperative to adapt SSMs to a discretized version to handle such input data effectively:

$$
\begin{aligned}
h_k &= \bar{A}h_{k-1} + \bar{B}x_k, \\
y_k &= \bar{C}h_k + \bar{D}x_k.
\end{aligned}
\tag{2}
$$

The discretized form of SSM utilizes a time-scale parameter $\Delta$ to transform continuous parameter $A$, $B$, $C$ and $D$ to discrete parameters $\bar{A}$, $\bar{B}$, $\bar{C}$ and $\bar{D}$. Especially, $\bar{D}$, which conventionally serves as a residual connection, is frequently simplified or omitted in certain contexts. The transition often uses the zero-order hold (ZOH)

discretization rule:

$$\bar{A} = (I - \Delta/2 \cdot A)^{-1}(I + \Delta/2 \cdot A),$$
$$\bar{B} = (I - \Delta/2 \cdot A)^{-1}\Delta B, \quad (3)$$
$$\bar{C} = C.$$

**Selective SSMs.** The inherent Linear Time-Invariant (LTI) characteristic of SSMs, which relies on the consistent utilization of matrices $\bar{A}$, $\bar{B}$, $\bar{C}$, and $\Delta$ across various inputs, imposes limitations on their ability to filter and comprehend contextual nuances within diverse input sequences. Mamba [13] address this limitation by treating $\bar{B}$, $\bar{C}$, and $\Delta$ as dynamic, input-dependent parameters, thereby transforming the SSM into a time-variant model. This modification enables the model to adapt more effectively to different input contexts, enhancing its capability to capture relevant temporal dynamics. Consequently, it obtains a more precise and efficient representation of the input sequence.

## 4 THE PROPOSED METHOD

### 4.1 Notation

As depicted in Figure 1, our proposed MambaTrack adheres to the tracking-by-detection paradigm [3, 6, 52] in an online manner. To this end, we employ an off-the-shelf detector, YOLOX [11], to acquire $M$ detections $\mathcal{B}_t = \{\mathbf{b}_i^t\}_{i=1}^M$ for the current frame $t$. Each detection $\mathbf{b} \in \mathbb{R}^4$ is represented by its 2D coordinates $(x, y)$ denoting the top-left bounding box corner in the image plane, alongside its width $w$ and height $h$. We denote the set of $N$ tracklets as $\mathbb{T} = \{\mathcal{T}_j\}_{j=1}^N$, where $\mathcal{T}_j = \{\mathbf{b}_j^s, \mathbf{b}_j^{s+1}, \cdots, \mathbf{b}_j^t\}$ denotes the tracklet of object $j$. Here, $\mathbf{b}_j^t$ signifies its bounding box in frame $t$, and $s$ denotes the frame of its initial appearance.

At the first frame of the video, we directly initialize the set of $\mathcal{T}$ with the detections $\mathcal{B}^1$. In subsequent frames, the goal is to assign the detection results provided by the detector to the appropriate tracklets. Over time, objects may exit the scene, leading to the termination of their trajectories and their subsequent removal from $\mathbb{T}$. Conversely, new objects may appear, and their trajectories will be added to $\mathbb{T}$. During tracking, trajectories are often interrupted due to occlusion and detector failure. Consequently, we further partition $\mathbb{T}$ into $\mathbb{T}_{active}$ and $\mathbb{T}_{lost}$, representing trajectories that have just been assigned new observations in the previous frame and trajectories that are temporarily interrupted but not yet removed, respectively. A tracklet $l$ in $\mathbb{T}_{active}$ is denoted as $\mathcal{T}_l = \{\mathbf{b}_l^s, \mathbf{b}_l^{s+1}, \cdots, \mathbf{b}_l^t\}$, while a tracklet $m$ in $\mathbb{T}_{lost}$ is represented as $\mathcal{T}_m = \{\mathbf{b}_m^s, \mathbf{b}_m^{s+1}, \cdots, \mathbf{p}_m^{t-2}, \mathbf{p}_m^{t-1}, \mathbf{p}_m^t\}$, where $\mathbf{b}$ denotes the bounding box provided by the detector, and $\mathbf{p}$ denotes the infilling bounding boxes used to fill the missing points caused by occlusion or detector failure.

### 4.2 Overview

The complexity of multiple object tracking in scenarios such as DanceTrack[40] and SportsMOT[50] arises from the intricate motion patterns of the objects and the substantial occlusion between them. To tackle this challenge, we adopt a divide-and-conquer framework to handle $\mathbb{T}_{active}$ and $\mathbb{T}_{lost}$ separately. First, we predict the spatial position of the active trajectories in the current frame based on their historical observations, utilizing the motion

predictor, Mamba Motion Predictor, proposed in this paper. Second, for the lost trajectories with missing observations, we employ autoregression to fill in the gaps before making predictions. We provide detailed explanations for each of these processes in the subsequent subsections.

### 4.3 Mamba Motion Predictor

An overview of the proposed Mamba Motion Prediction (MTP) is depicted in Figure 2, comprising three main components. The first component comprises an input embedding layer, which takes the historical dynamics of the object trajectory as input and linearly transforms it to obtain a sequence of input temporal tokens. The second component consists of an encoding layer composed of L bi-Mamba blocks with Mamba modules at its core. Finally, the last layer is the prediction head, responsible for predicting the inter-frame bounding box offsets of the object trajectory.

**Temporal Tokenization Layer.** For a tracklet $i$ in $\mathbb{T}$, we first construct the input trajectory feature:

$$\mathbf{O}_{in} = [\mathbf{o}_{t-q}, \mathbf{o}_{t-q+1}, \cdots, \mathbf{o}_{t-1}] \in \mathbb{R}^{q \times 4}, \quad (4)$$

where $q$ is the size of the look-back temporal window and $\mathbf{o} = [\delta c_x, \delta c_y, \delta w, \delta h]$, with $\delta c_x, \delta c_y, \delta w$, and $\delta h$ representing the normalized changes of the corresponding bounding box center, width, and height between two observation time steps. We utilize a single linear layer to obtain the input token sequence as follows:

$$\mathbf{X} = \text{Embedding}(\mathbf{O}_{in}), \quad (5)$$

where $\mathbf{X} \in \mathbb{R}^{q \times d_m}$ and $d_m$ is the dimension of the temporal token.

**Bi-Mamba Encoding Layer.** After obtaining the temporal tokens $\mathbf{X}$ of tracklets, we feed them into the designed bi-Mamba encoding layer to explore the motion patterns from the object's dynamic history. The bi-Mamba encoding layer comprises $L$ bi-Mamba blocks. Specifically, to fully utilize the information from the object trajectory and address the unidirectional limitation of Mamba, each bi-Mamba block contains bidirectional Mamba modules: one forward and one backward. For the $l$-th bi-Mamba block, the forward process can be formulated as follows:

$$\hat{\mathbf{X}}_{forward} = \text{Mamba}(\mathbf{X}_{l-1}),$$
$$\hat{\mathbf{X}}_{backward} = \text{Mamba}_{backward}(\mathbf{X}_{l-1}),$$
$$\hat{\mathbf{Y}} = \hat{\mathbf{X}}_{forward} + \hat{\mathbf{X}}_{backward}, \quad (6)$$
$$\mathbf{X}_l = \hat{\mathbf{Y}} + \text{LN}(\text{MLP}(\hat{\mathbf{Y}})),$$

where $\mathbf{X}_{l-1}$ is the output of the $(l-1)$-th bi-Mamba block, LN is the layer normalization function [1], and MLP is a two-layer multi-layer perceptron. The selective SSM is the core of the Mamba [13] module which is described in Sec. 3.

**Prediction head and training.** After being processed by the bi-Mamba encoding layer, an average pooling layer is utilized to aggregate the information from $\mathbf{X}_l$. Subsequently, a prediction head comprising two fully connected layers is employed to predict the offsets $\hat{\mathbf{O}}$. We utilize the smooth L1 loss to supervise the training process:

$$L(\hat{\mathbf{O}}, \mathbf{O}^*) = \frac{1}{4}\sum \text{smooth}_{L_1}(\hat{\delta}_i - \delta_i), i \in \{c_x, c_y, w, h\}, \quad (7)$$

where $\mathbf{O}^* = \{\delta_{c_x}, \delta_{c_y}, \delta_w, \delta_h\}$ represents the ground truth.

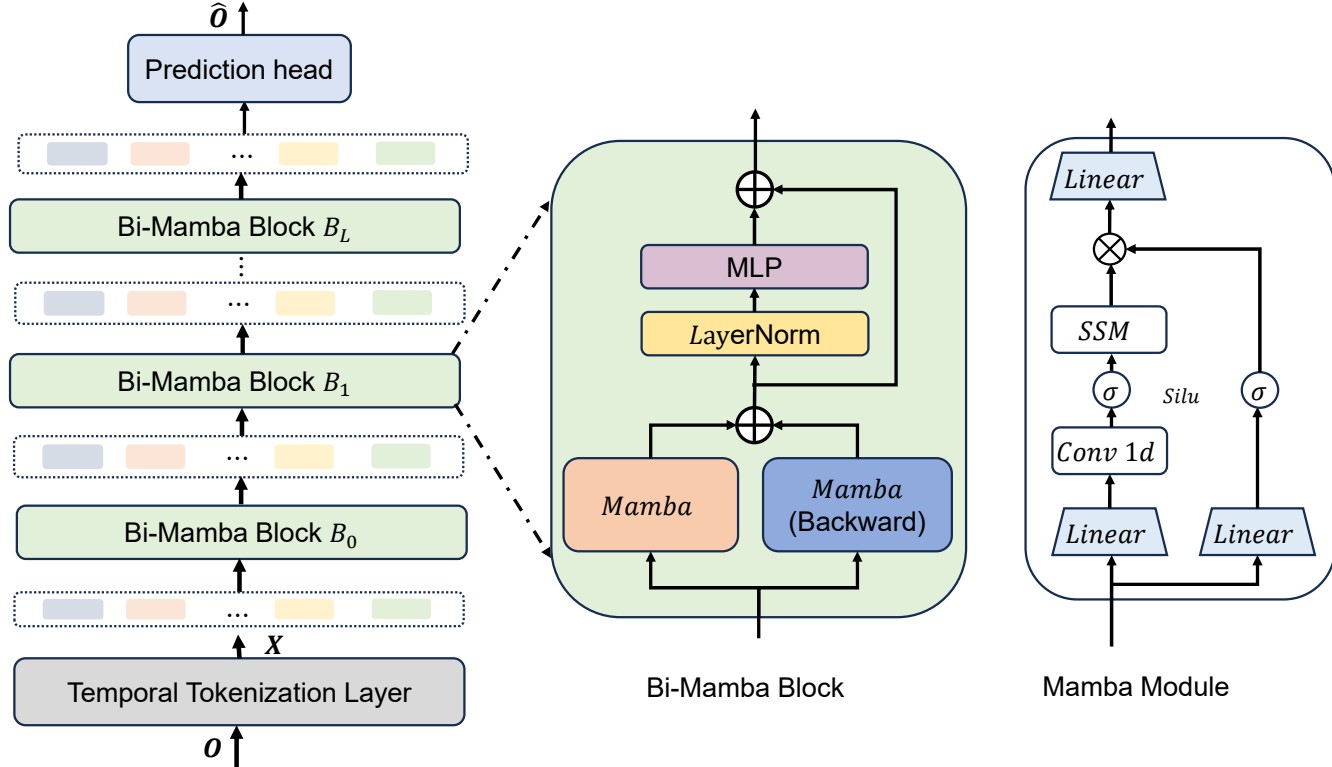

**Figure 2: Overview of the proposed Mamba motion predictor.**

## 4.4 Tracklet patching module

In real-world scenarios, objects may go undetected at certain time points due to severe occlusion or motion blur. Consequently, the corresponding tracklets may not receive new updates for several frames during the matching process, leading to early termination of the tracklets and fragmented trajectories. In this subsection, our goal is to extend the tracklets that do not receive new bounding boxes in order to enhance the consistency of the tracklets.

For example, if a lost tracklet $\mathcal{T}_i$ in lost tracklets $\mathbb{T}_{lost}$ receives no new update at the last time step $t-1$ and remains unmatched at the current frame $t$, we compensate for this missing observation in an autoregressive manner by considering the predicted bounding box $\hat{\mathbf{b}}_i^{t-1}$ as the actual observation of frame $t-1$. We then continue to predict its spatial location $\hat{\mathbf{p}}_i^t$ at the current frame. As shown in Figure 3, if it still fails to match with a new detection in the current frame, we persist in predicting its future bounding boxes frame by frame utilizing the motion predictor MTP, leveraging the historical trajectory sequence $\mathcal{T}_{past} = \{\cdots, \mathbf{b}_i^s, \mathbf{b}_i^{s+1}, \cdots, \hat{\mathbf{b}}_i^{t-1}\}$ and the predicted bounding box $\hat{\mathbf{p}}_i^t$ in an autoregressive manner:

$$\hat{\mathbf{p}}_i^{t+1} = \text{MTP}(\mathcal{T}_{past}, \hat{\mathbf{p}}_i^t). \tag{8}$$

Since the bounding boxes obtained through autoregression for lost tracklets are typically less reliable compared to those of active tracklets, we prioritize the association of active tracklets with the detection results $\hat{\mathcal{B}}_t$ in the current frame. Therefore, the active

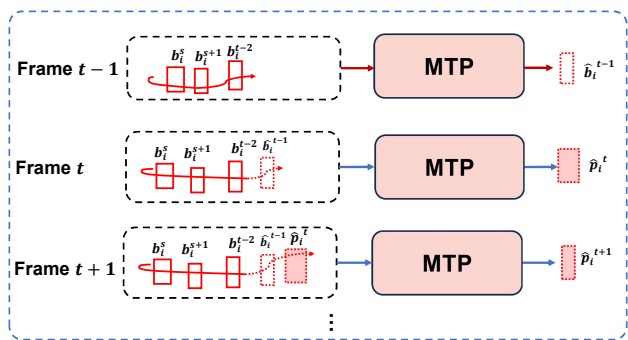

**Figure 3: In TPM, we utilize MTP in an autoregressive manner to extend the lost tracklets, providing an opportunity for their trajectories to be re-established in future frames.**

tracklets are given precedence in being associated with the detection results in the current frame. The remaining detection results are then associated with $\hat{\mathcal{P}}_t$, the predicted bounding boxes of the lost tracklets. The detailed inference process is described below.

## 4.5 Inference

During inference, we utilize the proposed Mamba motion predictor to model object motion patterns and predict their future movement.

**Algorithm 1:** Inference of MambaTrack at frame $t$.

**Input:** Detections: $\mathcal{B}_t = \{\mathbf{b}_i^t\}_{i=1}^M$, tracklets $\mathbb{T} = \{\mathcal{T}_j\}_{j=1}^N$ at frame $t-1$, Motion Predictor: MTP.

**Output:** Active tracklets $\mathbb{T}_{active}$ at current frame $t$.

    /* First Matching                                  */

1  $\mathbb{T}_{active}, \mathbb{T}_{lost} \leftarrow \mathbb{T}$

2  $\mathcal{B}_t \leftarrow [\mathbf{b}_t^1, \cdots, \mathbf{b}_t^{M_t}]$ // Detection set of current frame

3  $\hat{\mathcal{B}}_t \leftarrow [\hat{\mathbf{b}}_t^1, \cdots, \hat{\mathbf{b}}_t^N]$ from $\mathbb{T}_{active}$ // Predicted bounding boxes

4  $\mathbf{C}_t \leftarrow C_{\text{IoU}}(\hat{\mathcal{B}}_t, \mathcal{B}_t)$ // Cost matrix based on IoU similarity

5  $\mathcal{M}, \mathbb{T}_u, \mathcal{B}_u \leftarrow \text{Hungarian}(\mathbf{C}_t)$

6  $\mathbb{T}_{active} \leftarrow \{\mathcal{T}_i.update(\mathbf{b}_t^j), \forall (i,j) \in \mathcal{M}\}$

    /* Re-find lost tracklets via patched bounding boxes. */

7  $\mathbb{T}_{lost} \leftarrow \mathbb{T}_{lost} \cup \mathbb{T}_u$ // Lost tracklets

8  $\hat{\mathcal{P}} \leftarrow [\hat{\mathbf{p}}, \cdots, \hat{\mathbf{p}}_t]$ from $\mathbb{T}_{lost}$

9  $\mathbf{C}_{lost} \leftarrow C_{\text{IoU}}(\hat{\mathcal{P}}, \mathcal{B}_u)$

10  $\mathcal{M}, \mathbb{T}_u, \mathcal{B}_u \leftarrow \text{Hungarian}(\mathbf{C}_{lost})$

    /* Second Matching                              */

    /* Add the re-find lost tracklets to active tracklets */

11  $\mathbb{T}_{active} \leftarrow \{\mathcal{T}_i.update(\hat{\mathbf{p}}^j), \forall (i,j) \in \mathcal{M}\}$

    /* Update the lost tracklets with last predicted bounding boxes                   */

12  **for** $\mathcal{T}$ *in* $\mathbb{T}_{lost}$ **do**

13      |  $\mathcal{T}.update(\mathcal{T}.\hat{\mathbf{b}}_{t-1})$

14  **end**

15  $\mathbb{T} \leftarrow \mathbb{T}_{lost} \cup \mathbb{T}_{active}$

    /* Predict next bounding boxes of tracklets            */

16  **for** $\mathcal{T}$ *in* $\mathbb{T}$ **do**

17      |  $\text{MTP}(\mathcal{T})$

18  **end**

Following the common practice of SORT-like methods [3, 6], we implement the tracking process using bipartite matching, as depicted in Algorithm 1. We first associate the active tracks $\mathcal{T}_{alive}$ based on the Intersection-over-Union (IoU) similarities $\mathbf{C}_t$ between the predicted bounding boxes $\hat{\mathcal{B}}_t$ and detections $\hat{\mathcal{B}}_t$ in the current frame via the Hungarian algorithm [22]. Then, to find the lost tracklets, the remaining detections $\mathcal{B}_u$ will be matched with the predicted bounding boxes $\hat{\mathcal{P}}$ of them at the second matching based on $\mathbf{C}_{lost}$. Then, to find the lost tracklets, the remaining detections $\mathcal{B}_u$ will be matched with the patched bounding boxes $\hat{\mathcal{P}}$ in the second matching step.

For simplicity, we omit the initialization of new tracklets from the final remaining detection results $\mathcal{B}_u$ and the termination of lost tracks that have not received updates for consecutive $t_{\text{terminate}=30}$ frames. We initialize the unmatched detections whose confidence scores are higher than $t_{thresh} = 0.6$ as new tracklets.

## 5 EXPERIMENTS

### 5.1 Datasets and Metrics

**Datasets.** To assess the effectiveness of our proposed method, we conduct evaluations on two emerging datasets known for their diverse and rapid movements and indistinguishable appearances. These datasets are DanceTrack [40] and SportsMOT [9]. DanceTrack consists of 40 training videos, 25 validation videos, and 35 test videos. Objects in the dancing scenarios are easy to detect, but they are similar in appearance, difficult to distinguish, and exhibit complex and varied movement patterns. Additionally, the newly introduced SportsMOT dataset focuses on sports scenarios such as basketball, football, and volleyball. It contains 45 training videos, 45 validation videos, and 150 test video sequences collected from high-level sports events. Due to the fast and diverse motion of athletes, SportsMOT demands robust tracking approaches. The unique characteristics of the SportsMOT and DanceTrack datasets make them ideal benchmarks for evaluating motion-based trackers, avoiding shortcuts taken by appearance-based methods.

**Metrics.** To comprehensively evaluate the proposed algorithm, we employ a range of evaluation metrics, including the Higher Order Tracking Accuracy (HOTA), which encompasses Association Accuracy (AssA) and Detection Accuracy (DetA), as well as the IDF1 metric and metrics from the CLEAR family (MOTA, FP, FN, IDs, etc.) [27, 36, 37]. MOTA is computed from false negatives (FN), false positives (FP), and identity switches (IDs), and its calculation is primarily influenced by the quality of detection results. IDF1 primarily assesses the consistency of object trajectories. HOTA is specifically designed to provide a balanced assessment of both detection and association performance, making it the primary metric for evaluating tracker performance.

### 5.2 Implementation Details

This study focuses on developing a robust motion-based tracker, we utilize pre-trained weights of the YOLOX detector provided by the DanceTrack [40] and SportsMOT [9] benchmarks for fair comparisons. The bi-Mamba encoding layer comprises $L = 3$ bi-Mamba blocks with the input token dimension $d_m$ set to 512. The maximum look-back temporal window $q$ is set to 10, and the batch size is 64. We employ the Adam optimizer [21] with $\beta_1 = 0.9$, $\beta_2 = 0.98$, and $\epsilon = 10^{-8}$. During the training process, we adjust the learning rate according to the following formula to linearly increase the learning rate after $w_{\text{warmup}}$ training steps:

$$\text{lr} = (d_m)^{-0.5} \times \min(w^{-0.5}, w \times (w_{\text{warmup}})^{-1.5}), \qquad (9)$$

where $w$ is the training step number, and we set the $w_{\text{warmup}} = 4000$.

### 5.3 Benchmark Results

**DanceTrack.** As presented in Table 1, our proposed MambaTrack outperforms state-of-the-art methods in the key metric HOTA, demonstrating a lead of 2.2 percentage points over OC_SORT without any postprocessing. OC_SORT addresses the limitations of the Kalman Filter in handling nonlinear motion and heavily occluded environments. Our tracker is designed to model the diverse motion patterns of objects and enhance robustness against short-term missing observations. This is corroborated by achieving the highest IDF1 score of 57.8, surpassing the second-best method by 3.2 percentage points. Furthermore, our method demonstrates improved accuracy in predicting future spatial locations of objects compared to motion-based trackers [3, 6, 52] employing the same

**Table 1: Evaluation on on DanceTrack test set. The best results are shown in bold. Values that are higher or lower, marked by ↑ /↓, are indicative of better performance.**

| Tracker | IoU | Appear. | HOTA↑ | IDF1↑ | AssA↑ | MOTA↑ | DetA↑ |
|---|---|---|---|---|---|---|---|
| DeepSORT [46] | ✓ | ✓ | 45.6 | 47.9 | 29.7 | 87.8 | 71.0 |
| MOTR [51] | ✓ | ✓ | 54.2 | 51.5 | 40.2 | 79.7 | 73.5 |
| FairMOT [53] | ✓ | ✓ | 39.7 | 40.8 | 23.8 | 82.2 | 66.7 |
| TransTrk [41] | ✓ | ✓ | 45.5 | 45.2 | 27.5 | 88.4 | 75.9 |
| TraDes [47] | ✓ | ✓ | 43.3 | 41.2 | 25.4 | 86.2 | 74.5 |
| QDTrack [32] | | ✓ | 45.7 | 44.8 | 29.2 | 83.0 | 72.1 |
| CenterTrack [54] | ✓ | | 41.8 | 35.7 | 22.6 | 86.8 | 78.1 |
| SORT [3] | ✓ | | 47.9 | 50.8 | 31.2 | **91.8** | 72.0 |
| ByteTrack [52] | ✓ | | 47.3 | 52.5 | 31.4 | 89.5 | 71.6 |
| OC_SORT [6] | ✓ | | 54.6 | 54.6 | **40.2** | 89.6 | **80.4** |
| Ours | ✓ | | **56.8** | **57.8** | 39.8 | 90.1 | 80.1 |

**Table 2: Evaluation on SportsMOT test set. The best results are shown in bold. Values that are higher or lower, marked by ↑ /↓, are indicative of better performance.**

| Tracker | Motion | Appear. | HOTA↑ | IDF1↑ | AssA↑ | MOTA↑ | DetA↑ |
|---|---|---|---|---|---|---|---|
| FairMOT[53] | ✓ | ✓ | 49.3 | 53.5 | 34.7 | 86.4 | 70.2 |
| MixSort-Byte [9] | ✓ | ✓ | 65.7 | 74.1 | 54.8 | 96.2 | 78.8 |
| MixSort-OC [9] | ✓ | ✓ | 74.1 | 74.4 | 62.0 | 96.5 | 88.5 |
| TransTrack[41] | ✓ | ✓ | 68.9 | 71.5 | 57.5 | 92.6 | 82.7 |
| GTR[56] | | ✓ | 54.5 | 55.8 | 45.9 | 67.9 | 64.8 |
| QDTrack[32] | | ✓ | 60.4 | 62.3 | 47.2 | 90.1 | 77.5 |
| CenterTrack[54] | ✓ | | 62.7 | 60.0 | 48.0 | 90.8 | 82.1 |
| ByteTrack[52] | ✓ | | 62.8 | 69.8 | 51.2 | 94.1 | 77.1 |
| OC-SORT[6] | ✓ | | 71.9 | 72.2 | 59.8 | 94.5 | 86.4 |
| Ours | ✓ | | **72.6** | **72.8** | **60.3** | 95.3 | **87.6** |

detector. Furthermore, our approach outperforms methods that exploit appearance information. This underscores the significance of utilizing motion information, particularly in complex scenarios like DanceTrack, characterized by intricate object motion patterns and homogeneous appearances.

**SportsMOT.** As shown in Table 2, our proposed tracker, MambaTrack, outperforms comparable tracking algorithms that rely solely on motion information across all metrics. Notably, our method exhibits a substantial lead over ByteTrack, which utilizes Kalman Filter, by nearly 10 percentage points in the HOTA metric, and by 3 percentage points and 9.1 percentage points in the IDF1 and AssA metrics, respectively, which assess trajectory consistency. Additionally, our method surpasses OCSORT, an enhanced Kalman Filter-based approach, demonstrating superior performance. These results underscore the advanced capabilities of our method, even in challenging scenarios characterized by the fast and diverse movements of athletes, further validating its effectiveness.

## 5.4 Ablation Study

In this section, we perform ablation experiments to validate the effectiveness of our proposed Mamba Motion Predictor (MTP) and the tracklet patching module (TPM). All models are trained on the

**Table 3: Ablation studies on Mamba Motion Predictor (MTP) and Tracklet Patching Module (TPM).**

| | HOTA↑ | IDF1↑ | AssA↑ | MOTA↑ | DetA↑ |
|---|---|---|---|---|---|
| baseline | 45.9 | 50.9 | 30.7 | 86.3 | 69.0 |
| + MTP | 54.9 | 54.5 | 38.5 | **89.3** | **78.6** |
| + TPM | **55.1** | **56.1** | **39.2** | 89.01 | 77.7 |

DanceTrack [40] dataset and evaluated on the DanceTrack validation set. We implement a baseline utilizing the Kalman Filter as the motion predictor.

**Effectiveness of the proposed MTP and TPM.** As depicted in Table 3, we evaluate the contributions of the proposed modules. It is evident from the table that our proposed motion predictor has led to significant improvements across all metrics compared to the baseline method. Specifically, HOTA demonstrates a 9 percentage point improvement, while IDF1 and AssA show enhancements of 3.6 and 7.8 percentage points, respectively. This underscores the effectiveness of the MTP in efficiently modeling the nonlinear motion of objects and accurately predicting their positions in adjacent frames compared to the Kalman Filter. Furthermore, the introduction of the TPM module results in additional enhancements in metrics related to trajectory consistency, with IDF1 and AssA improving by 1.6 and 0.7 percentage points, respectively.

**Table 4: Comparison of different motion models.**

|          | HOTA↑ | IDF1↑ | AssA↑ | MOTA↑ | DetA↑ |
|----------|-------|-------|-------|-------|-------|
| None(IoU) | 44.7 | 36.8 | 25.3 | 87.3 | **79.6** |
| KF       | 45.9 | 50.9 | 30.7 | 86.3 | 69.0 |
| LSTM     | 51.3 | 51.6 | 34.4 | 87.1 | 76.7 |
| TF       | 52.5 | 52.5 | 35.2 | **89.3** | 78.5 |
| MTP      | **54.9** | **54.5** | **38.5** | **89.3** | 78.6 |

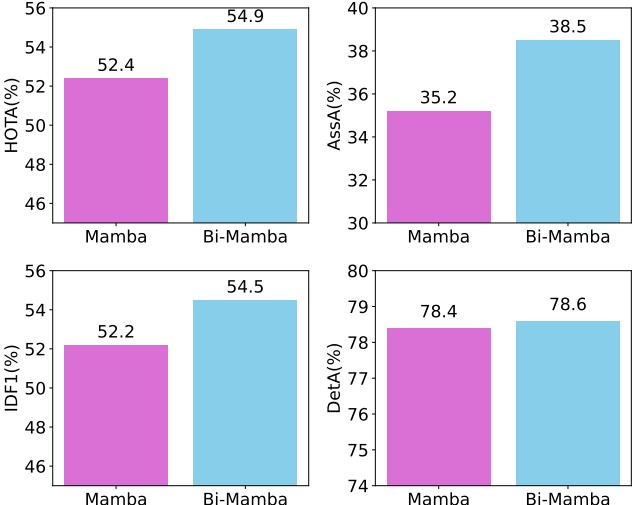

**Figure 4: Performance comparison between the proposed bi-Mamba block and the vanilla Mamba.**

**Table 5: Impact of different numbers $L$ of Bi-Mamba blocks.**

| $L$ | HOTA↑ | IDF1↑ | AssA↑ | MOTA↑ | DetA↑ |
|-----|-------|-------|-------|-------|-------|
| 1   | 52.1 | 51.8 | 34.7 | 89.2 | 78.5 |
| 2   | 54.1 | 54.4 | 37.4 | **89.3** | **78.7** |
| 3   | **54.9** | **54.5** | **38.5** | **89.3** | 78.6 |
| 4   | 52.1 | 52.1 | 34.7 | **89.3** | 78.5 |
| 5   | 52.3 | 52.4 | 35.1 | **89.3** | 78.3 |

**Different motion modeling.** We conduct a comparative analysis to assess the impact of temporal dynamic information provided by different motion models on data association, as summarized in Table 4. Across all motion models, significant improvements are observed compared to the most basic approach, which solely relies on IoU matching without incorporating motion information. Specifically, all the data-driven motion models, which leverage identical trajectory features, demonstrate superior performance compared to the Kalman Filter [20]. Notably, in terms of the HOTA metric, LSTM [8] exhibits a 5.4 percentage point improvement, Transformer (TF) [42] leads by 6.6 percentage points, while our proposed MTP achieves the highest improvement of 9 percentage points. These findings emphasize the substantial potential of data-driven motion-based models and affirm the efficacy of our proposed SSM-based MTP module.

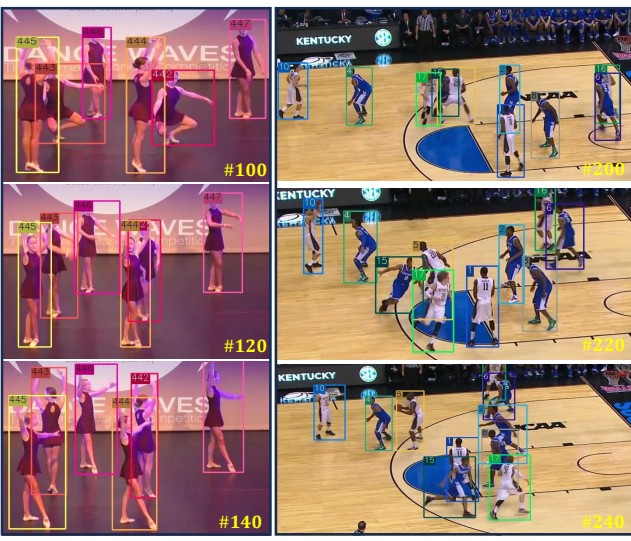

**Figure 5: Qualitative results on DanceTrack [40] and SportsMOT [50].**

**Ablation on the design of bi-Mamba encoding layer.** We conduct further analysis on the design of the bi-Mamba encoding layer. As depicted in Figure 4, Bi-Mamba exhibits superior performance across various metrics including HOTA, IDF1 and AssA. Specifically, it leads by 2.5 percentage points in the HOTA metric and demonstrates improvements of 3.3 and 2.3 percentage points in the AssA and IDF1 metrics, respectively. These results confirm the effectiveness of utilizing Bi-Mamba in capturing the object's motion patterns and enhancing the accuracy of object motion prediction compared to using the vanilla Mamba [13] only. Furthermore, we evaluate the impact of different numbers of bi-Mamba blocks in Table 5. Setting $L$ to 3 yields the optimal performance for MTP, achieving the highest values for HOTA, IDF1, AssA, and MOTA metrics.

**Qualitative Analysis.** As depicted in Figure 5, our proposed MambaTrack accurately tracks objects in randomly selected videos featuring crowded group dancing and basketball games, where objects move rapidly and change direction irregularly.

## 6 CONCLUSION

This paper introduces an online motion-based tracker comprising a motion predictor and a tracklets patching module. The Mamba Motion Predictor, grounded in the State Space Model, Mamba, effectively models the temporal dynamics of objects, facilitating accurate association between objects in consecutive frames. Besides, to enhance trajectory consistency, we leverage the motion predictor as an autoregressor to predict bounding boxes for lost trajectories, thereby re-establishing them. Despite its simplicity and intuitiveness, experimental results on complex motion datasets validate the effectiveness of our approach. We aim for our proposed method to serve as a baseline, fostering further exploration and development of motion-based tracking algorithms.

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
