# OpenReview forum: "MambaTrack: a simple baseline for multiple object tracking with State Space Model"
_acmmm.org/ACMMM/2024/Conference — MM2024 Oral_

### Official Review · Reviewer_TbeB · 2024-05-17

**Rating:** 5
**Confidence:** 3

**Summary:**

This paper proposes an online motion-based multi-object tracking method by employing a Mamba motion predictor. The motivation is to improve tracking performance in complex scenarios when the objects exhibit nonlinear and diverse motion patterns. The proposed method is evaluated on two challenging datasets: DanceTrack and SportsMOT. Experimental results show that the proposed approach achieves robust tracking performance compared with the baseline models. Ablation studies has been conducted to help understand the effectiveness of different modules of the MambaTrack model.

**Strengths:**

- The paper is well-written and easy to follow.

- The proposed approach is technically sound and reasonable.

- Experimental results provide evidence that the proposed approach exhibits superior performance compared to other baseline MOT methods.

- The ablation studies are sufficient to demonstrate the effectiveness of the different modules in the proposed model.

**Limitations:**

Several expressions in the paper are inaccurate and discussable.

- Section 3, line 286. "State Space Models (SSMs) are inspired by linear time-invariant (LTR) system..." In fact, the SSM is a general term and not limited to linear systems. It would be better to clarify that SSMs are general mathematical frameworks to describe a dynamical system and LTI systems are a specific case of SSMs. By the way, there is a typo for "LTR" -> "LTI"

- Section 3, line 334. "Since SSMs are designed to operate on continuous signal x(t), they inherently cannot process discrete sequences..." Again, the SSMs are not limited to continuous-time signals, they can have both continuous-time formulation and discret-time formulation. It is inaccurate to say, "they inherently cannot process discrete sequences".

Several technical details are missing.

- If I understood correctly, the size of the so-called "look-back temporal window" is fixed. Please clarify in Section 5.1 how the video sequences are processed. Are they split into fixed-size sequences for training? Besides, during inference, when t<q (i.e. for time frames shorter than the window size), how the input sequence are processed? Are they padded with specific values?

- Please explain why the best scores reported in Table 3 does not correspond to that in Table 1.

Typos

- Section 4.5, line 620, there is a typo for the notation of detections.

- Section 4.5, line 624. The sentence is redundant with the previous one.

Further improvements

- It would be better to point out that the scores reported in Table 1 and Table 2 for the baseline models are reproduced or directly taken from the corresponding papers.

- Section 5.4, for the qualitative analysis. It would be more convincing to show examples of how the proposed model can address challenging tracking scenarios with occlusions, accompanied by detailed explanations for specific objects.

**Suitability:**

2

---

### Official Review · Reviewer_smpj · 2024-05-24

**Rating:** 4
**Confidence:** 3

**Summary:**

This paper introduces MambaTrack, an advanced tracking system utilizing a novel Mamba-based motion model named Mamba moTion Predictor (MTP) to enhance Multi-object Tracking (MOT) capabilities. It employs bi-Mamba encoding layers to capture complex motion patterns from the spatial-temporal dynamics of objects, enhancing prediction accuracy. Furthermore, MTP is applied in an autoregressive manner to handle instances of occlusions or motion blur by using its predictions as inputs for subsequent predictions, thus ensuring more consistent trajectories.

**Strengths:**

1.The integration of the Mamba-based State Space Model (Mamba moTion Predictor, MTP) in multi-object tracking (MOT) presents a novel approach, particularly in handling complex motion scenarios like dancing and sports. The utilization of a bi-Mamba encoding layer to capture the dynamic motion patterns is an innovative technical advancement that improves tracking accuracy in challenging environments.
2.While the paper discusses the near-linear complexity of the Mamba model during inference, detailed computational costs, including time and resource requirements for real-time applications, are not thoroughly addressed. This oversight might concern potential adopters of the technology in resource-constrained environments.
3.The paper leverages the strengths of state space models, known for their efficiency in handling sequential data, to address the specific challenges in MOT, offering a theoretical justification for their use in predicting object motion over time, which is both innovative and grounded in solid mathematical frameworks.

**Limitations:**

1.The performance of MambaTrack heavily relies on the accuracy of the initial detections provided by detectors like YOLOX. This dependence could limit its effectiveness in scenarios where detector performance is suboptimal due to poor lighting conditions, extreme weather, or other adverse environmental factors.
2.The association between remaining detections and lost tracklets is described twice in the first paragraph of Section 4.5. This repetition could potentially confuse readers and detract from the manuscript's overall clarity. It is recommended that the authors carefully review and edit the manuscript to eliminate such redundancies before submission.
3.By substituting the conventional motion predictors with the Mamba Motion Predictor (MTP) in a variety of existing tracker and conducting a series of experiments, the effectiveness and superiority of MTP can be more comprehensively validated.

**Suitability:**

3

---

### Official Review · Reviewer_eDB6 · 2024-05-24

**Rating:** 4
**Confidence:** 3

**Summary:**

To handle the non-linear motion modeling problem in MOT, this paper proposes to use a Mamba-based learnable motion predictor named MTP instead of the traditional way using KF. Experiments show that MTP has better motion modeling capability than KF in complex motion scenarios. By combining MTP and a simple TBD tracking pipeline, MambaTrack achieves good results on DanceTrack and SPORTSMOT.

Generally, the idea of using Mamba in MOT is novel and meaningful, and the writing of this manuscript is clear.

**Strengths:**

1. The authors propose the Mamba-based MPT module to replace the KF for motion prediction and demonstrate its effectiveness through experiments. Such an attempt demonstrates the feasibility of using Mamba for improving MOT tracking performance.
2. A TPM is also proposed to continuously predict motion states of lost tracklets using previous observations and predictions, which also brings performance gains.
3. This work also inspires more work to explore more effective and learnable motion models in MOT instead of KF to deal with complex motions.

**Limitations:**

1. For the TPM module, a concern is that will the continuous prediction of lost trajectories based on historical observations and predictions lead to error accumulation similar to the KF prediction without update?
2. In the first paragraph of Section 4.5, the author describes the association between remaining detections and lost tracklets twice. The authors should proofread the manuscript before submission to reduce such mistakes.
3. In Table 3, is the baseline with the same tracking flow as MambaTrack but using KF as the motion predictor? The authors can also try to replace the KF in other trackers with MTP to further demonstrate its effectiveness.

**Suitability:**

2

---

### Official Review · Reviewer_rC96 · 2024-05-25

**Rating:** 3
**Confidence:** 3

**Summary:**

This paper integrates the famous Mamba to object tracking. To address the nonlinear and diverse motion in special scenarios, the authors resort to exploring data-driven motion prediction methods and propose Mamba moTion Predictor (MTP). MTP takes the spatial-temporal location dynamics of objects as input, captures the motion pattern using a bi-Mamba encoding layer, and predicts the next motion. The proposed tracker, MambaTrack, demonstrates advanced performance on benchmarks such as Dancetrack and SportsMOT

**Strengths:**

This paper integrates Mamba to object tracking and achieves good results on DanceTrack and SportsMOT.
1. This paper is easy to follow.
2. The writing of this paper is good.
3. The experimental results are relatively outstanding.

**Limitations:**

The biggest drawback of this paper is that MambaTrack is only an integration not an innovative idea.
Of course, Mamba has been a hot topic in recent months. But for object tracking, an attractive paper is not just an integration of some new methods. The most important problems of object tracking such as occlusion and data association are still solved in traditional ways or not mentioned. The proposed MTP is designed to predict the complex motion of objects. However, I have studied many studies that claim they try to solve nonlinear motion patterns, but no one has solved this problem actually.
For this paper, MTP only takes sequential motion information as input, and predicts the final offsets. This is a comment way in many papers [1][2] and so on. Maybe others use LSTM or Transformer, but this paper takes account of Mamba. However, the motion information still follows the same processing way.
The TPM is similar to the recurrent predicting pattern in the Kalman filter or other RNN-based methods.
Finally, if the authors want to demonstrate the advantages of MambaTrack. At least, they should analyze or compare mamba-based motion prediction with RNN/Transformer-based prediction. Then the reviewers can figure out why the authors select Mamba.
[1] Mamba-FETrack: Frame-Event Tracking via State Space Model, 2024.
[2]  Exploring Learning-based Motion Models in Multi-Object Tracking, 2024.

**Suitability:**

1

---

### Meta-Review · Area_Chair_sAr4 · 2024-07-03

**Recommendation:** Accept (Oral)
**Confidence:** 5

**Metareview:**

This paper proposes a MambaTrack integrating Mamba-based motion prediction to enhance Multi-object Tracking capabilities. The paper is well-written and easy to follow, presenting a clear description of the proposed method and its experimental validation. The proposed Mamba moTion Predictor (MTP) utilizes bi-Mamba encoding layers to predict complex object motions, demonstrating promising results on benchmarks such as DanceTrack and SportsMOT. The integration of Mamba into MOT via MambaTrack, particularly through MTP, represents a advancement in handling nonlinear and diverse motion patterns. The use of bi-Mamba encoding layers for motion prediction shows promising results, enhancing tracking accuracy in difficult scenarios. While the paper has some novelty, several key issues need to be addressed to improve its technical strength: 1) theoretical grounding of MambaTRack's advantages when compared with SOTA architectures, 2) more ablation studies and a more thorough comparison with other tracking methods using alternative motion predictors, 3)  analysis of computational cost and real-time feasibility.